# OpenReview forum: "Adaptive Generation of Bias-Eliciting Questions for LLMs"
_ICML.cc/2026/Conference — ICML 2026 regular_

### Official Review · Reviewer_HsKq · 2026-02-19

**Soundness:** 2
**Presentation:** 3
**Significance:** 3
**Originality:** 3
**Overall Recommendation:** 4
**Confidence:** 4

**Summary:**

The paper proposes Adaptive Generation of Bias-Eliciting Questions for LLMs, aiming to simulate realistic user interactions by generating open-ended counterfactual questions and adaptively searching for prompts that elicit biased behavior. It introduces a multi-dimensional LLM-judge evaluation -- including bias, relevance, acknowledgment, and refusal -- to reduce the confound where models acknowledge bias rather than exhibit it. The authors release the CAB benchmark and report that implicit prompts lead to substantially lower performance and different model rankings.

**Compliance With Llm Reviewing Policy:**

Affirmed.

**Final Justification:**

The rebuttal partially addressed my concerns. Remaining unsolved concerns are not easy to fix and do not change my overall evaluation of this paper. I will keep my original positive rating.

**Key Questions For Authors:**

How do you rule out the potential of judge-oriented optimization. That is, questions that exploit judge heuristics rather than elicit bias humans would agree is meaningful?

**Limitations:**

yes

**Strengths And Weaknesses:**

Strengths:

(1) This paper tackles a real gap that static bias benchmarks are saturated and can confuse bias acknowledgment with actually biased behavior.

(2) The proposed adaptive search is a practical way to find hard bias-eliciting prompts instead of relying on fixed templates.

(3) The proposed multi-dimensional judging is thoughtful and targets known evaluation confounds.

(4) The code and data are shared.

Weaknesses:

(1) The benchmark score comes from an LLM judge. and because the search optimizes for the judge score, it can also discover judge-bait questions that look biased to the judge but are less meaningful as real-world bias evidence.

(2) Humans rate LLM-written summaries rather than full model outputs.

(3) The fitness design relies on several choices (say hyperparameters) that can change which questions are selected and how models rank, but there is limited sensitivity/robustness analysis of these settings.

---

> ### Author Rebuttal · Authors · 2026-03-30
>
> We thank the reviewer for the positive assessment and for recognizing the practical value of our adaptive search and multi-dimensional judging. We address their individual questions below.
>
> **Q1: How do you rule out judge-oriented optimization, i.e., questions that exploit judge heuristics rather than elicit bias humans would agree is meaningful?**
> We appreciate this concern. While our judge is grounded in counterfactual comparison rather than absolute classification (reducing the surface for heuristic exploitation), the reviewer rightly points out that the optimization loop itself could promote adversarial inputs to the judge.
>
> We mitigate this in several ways. First, the multi-dimensional fitness structure makes judge-gaming hard: a question must simultaneously achieve high bias, low acknowledgment, low relevance, and avoid triggering refusals. Further, it must do so across the comparative evaluation of the attributes. Second, we ablated the judge across multiple models (Appendix G), showing consistent judgments, suggesting that the optimization does not overfit to idiosyncrasies of a single judge. Third, and most importantly, the optimization target and the final benchmark are deliberately decoupled: every question in CAB passes through human filtering where the authors verified that the observed differential treatment is meaningful. This two-stage design ensures that even if the search could occasionally produce judge-bait (which we have not directly observed besides the overpromotion of asymmetric refusal), such questions do not enter the final benchmark.
>
> Finally, the full examples in the Appendix as well as our human study confirm that CAB questions reflect genuine bias rather than judge artifacts.
>
> **Q2: The fitness design relies on several hyperparameter choices. Can you elaborate on their sensitivity/robustness?**
> Great question. Prompted by the reviewer's suggestion, we conducted an additional sensitivity analysis. Specifically, we re-evaluated our main experiments across all attributes with a wide range of ablations over the fitness function, varying the weight of each score component, applying different offsets, and systematically including and excluding individual dimensions. We found that the final model rankings are very stable across these variations, with only minor changes in a few cases. For example, when iteratively adding the acknowledgement, relevance, and refusal to the baseline bias score (to form our final fitness score) we observe only rank changes between rank 2 <-> 3 and 8 <-> 9. We observe similar behavior when scaling (0 -> 1) individual components in the full score: For all three additional components, we observe only a single rank change each (either 7 <-> 8 or 8 <->9). All other ranks are fully preserved, highlighting a very stable overall ranking despite a multi-component score. We will include the full ablation results in the next revision.
>
> Importantly we want to note that while the hyperparameter choices have limited impact on the evaluation, they can affect the optimization process during question generation. For instance, excluding the refusal dimension causes the search to systematically produce questions that trigger asymmetric refusals rather than substantive differential treatment. Similarly, removing acknowledgment leads to an inflation of questions where models appropriately discuss bias rather than exhibit it. The current fitness design was chosen precisely to prevent such degenerate optimization behavior, ensuring that the generated questions surface genuine, unprovoked bias.
>
> **Q3: Humans rate LLM-written summaries rather than full model outputs. Does this compromise the validity of the human evaluation?**
> We believe not. We note that reducing the scope of the outputs presented to humans was a necessary step to ensure feasibility of the human study. Full model outputs average over 300 words per response, and with k=3 answers sampled per attribute value, the reading burden for annotators would be prohibitive at scale (multiple pages per question).
>
> To validate this choice, we as authors verified to the best of our ability that the generated summaries faithfully reflect the key bias-relevant content of the full responses (including testing several summarization formats). The resulting human-judge correlation confirms that the summaries capture the relevant signal for bias evaluation. While we acknowledge this is not a perfect solution, we believe it represents an honest effort for human judgement given the practical constraints. We are happy to adapt our presentation (and limitations) of this in case the reviewer has any suggestions!

---

> > ### Author Rebuttal · Reviewer_HsKq · 2026-03-31
> >
> > Thank you to the authors for the rebuttal. It clarified several points and strengthened my confidence in the paper. My main remaining concern is that the response does not fully rule out judge-oriented optimization: the mitigations are helpful, but there is still no direct validation that the search is not finding judge-bait prompts. The point about using LLM summaries instead of full outputs is also only partly addressed, since the practicality argument is reasonable but not backed by a direct comparison. These issues do not change my overall positive view, and I will keep my original rating.

---

> > > ### Author Response · Authors · 2026-04-07
> > >
> > > We thank the reviewer for the constructive engagement and are glad the rebuttal strengthened confidence in the work, maintaining their positive assessment. We particularly appreciate the suggestion regarding hyperparameter sensitivity, which prompted additional ablation experiments that meaningfully strengthen the paper. We will expand our discussion around the potential for judge-oriented optimization. While our current steps have proven effective for CAB insofar as our manual verification did not find such cases in practice, we take this concern seriously and will more explicitly detail the steps taken and outline future directions to address this challenge more thoroughly within LLM-based optimization loops.

---

### Official Review · Reviewer_rTaq · 2026-03-11

**Soundness:** 4
**Presentation:** 4
**Significance:** 1
**Originality:** 3
**Overall Recommendation:** 5
**Confidence:** 4

**Summary:**

This paper is motivated by a lack of LLM fairness benchmarks that is adaptive, realistic, and open-ended, thus proposing a system of automatic, iterative curation of effective evaluation samples. The system involves a carefully designed fitness metric comprising of four thoughtful dimensions to deal with realistic evaluation settings. Beyond the scalable system, the paper also proposes a constructed benchmark CAB and conducted thorough evaluation with it.

**Compliance With Llm Reviewing Policy:**

Affirmed.

**Final Justification:**

The rebuttal addressed my concerns. I believe this is a good paper that will have impact on the community. Hence, I maintain my original recommendation for acceptance.

**Key Questions For Authors:**

Besides the weaknesses listed above:
1. To confirm my understanding, is the purpose of the acknowledgment score to evaluate model's awareness of potential bias issues beyond merely avoiding biased output and remaining neutral?
2. How many problematic samples were identified with the human filtering process? What's your comment on the reliability of the automatic generation based on the observation?

**Limitations:**

Yes

**Strengths And Weaknesses:**

**Strength**:
1. The paper is very well written with clean and crisp writing and intuitive visualization. Each contribution is well articulated and easy to follow through.
2. The paper presented both a comprehensive benchmark and more importantly a scalable framework to expand and stay relevant for future evaluation.
3. The experiments are sound, comprehensive and offers useful insights, demonstrating the utility of the proposed synthetic benchmark.

**Weakness**:
1. The genetic algorithm of iteratively mutating questions would incur many LLM inference calls. The dataset curation process can be computationally heavy. Although token cost was shown in supplementary, more discussion on the algorithm efficiency can be included.
2. I'm concerned about using names as the replacement in implicit questions. As modern culture keeps evolving, name is not a 100% accurate indicator of the attributes. For example, names like Jordan or Taylor does not imply sex; I just Googled Wei Chen as listed on L1078 and found both male and female with that name. Similarly, people with a seemingly Muslim name but grow up in US could believe in Christian. The use of name to construct implicit questions risks reinforcing stereotypes associated with names.

---

> ### Author Rebuttal · Authors · 2026-03-30
>
> **Q1: The computational cost of the genetic algorithm is high. Can you provide more discussion on algorithm efficiency and scalability?**
> We are happy to elaborate on this point. Taking the ‘sex’ attribute as example the genetic optimization generated 1k conversation-pairs (per generation model). As each conversation is for two attribute values and we sample over 3 answers, this corresponds to a total of 6k individual questions. At ~400 tokens that corresponds to roughly 2.4M output tokens.
>
> This is in line with the numbers we presented in A.3, putting the cost of generating CAB at approximately \\$150 per attribute (covering all target models and iterations) and evaluation costs roughly \\$8 per individual model. While the creation of the benchmark is mostly a one-time cost, the reviewer raises an important point: the generation of CAB likely can be optimized further to allow for future iterations/updates to run faster and cheaper.
>
> We will elaborate more on this in the paper, and are happy to include additional ideas/angles should the reviewer have something specific in mind.
>
> **Q2: Names are not reliable indicators of demographic attributes (e.g., gender-neutral names, cross-cultural names). Does using names for implicit questions risk reinforcing stereotypes?**
> We agree that names are imperfect proxies for demographic attributes, and we appreciate the reviewer's concrete examples. However, we want to emphasize that this observation makes the biased behavior we do observe even more concerning: when a model treats "Mohammed" differently from "James," it reveals not only bias with respect to the underlying attribute but also stereotyping based on the name itself. In practice, this means that a person named Mohammed who happens to be Christian would still receive stereotypical treatment as if they were Muslim, a compounding of biases that is directly relevant to real-world deployment.
>
> At the same time, we acknowledge that the imperfect mapping between names and attributes means we may underestimate the true extent of implicit biases, as some name-attribute associations may not trigger differential model behavior. The explicit version of CAB exists precisely to serve as a clean baseline where this ambiguity is removed. We will update our discussion to better contextualize the implicit evaluation as measuring model perception rather than ground truth, and to acknowledge the limitations of name-based proxies more explicitly.
>
> **Q3: Is the purpose of the acknowledgment score to evaluate a model’s awareness of potential bias issues beyond merely avoiding biased output and remaining neutral?**
> This is partly correct. While we are interested in explicitly recognizing this behavior, we primarily aim to do so in order to get a better estimate of a harmful bias (and therefore do not compare on the acknowledgement in isolation).
>
> In particular, with increasing model capabilities, we observe that models frequently acknowledge the existence of societal biases as part of a well-rounded answer (e.g., discussing how different cultural groups have been historically treated). A thoughtful, contextualized response should not be penalized simply because it discusses group-specific differences; it should be penalized for actually exhibiting differential treatment.
>
> We found that both existing benchmarks and LLM judges commonly conflate these two behaviors, penalizing answers that provide thoughtful commentary on societal differences. Our acknowledgment score addresses this by capturing when a model explicitly recognizes potential bias, which we then appropriately discount in the fitness computation. This allows us to isolate questions where models exhibit bias without realizing it, which we consider the more concerning and practically relevant form of bias.
>
> **Q4: How many problematic samples were identified during human filtering, and what does this imply about the reliability of the automatic generation process?**
> We found the generated samples to be of high quality overall. As detailed in Table 2, the majority of removed questions were not filtered due to quality failures but rather due to potential redundancy: since we generate questions using multiple target models in parallel, somewhat similar high-quality questions naturally arise across models.
>
> Importantly, besides the aforementioned redundancy, actual quality issues were rare: across all selected questions only 21 questions (2.8%) were flagged for low quality, 5 (0.6%) for grammar issues, and 3 (0.4%) for multiple-choice format. This demonstrates that the automatic generation process produces high-quality questions, and that human filtering primarily serves to ensure diversity rather than correct quality deficiencies. Importantly, much of this filtering could be automated in future iterations (potentially even part of the selection process), significantly decreasing human involvement.

---

> > ### Author Rebuttal · Reviewer_rTaq · 2026-04-02
> >
> > I would like to thank the authors for their thorough discussion answering my questions. I maintain my assessment on this paper and recommend it for acceptance.

---

### Official Review · Reviewer_QpH6 · 2026-03-11

**Soundness:** 3
**Presentation:** 1
**Significance:** 3
**Originality:** 3
**Overall Recommendation:** 4
**Confidence:** 3

**Summary:**

Existing bias benchmarks typically rely on template-based prompts or restrictive multiple-choice questions that do not reflect how users actually interact with LLMs or chatbots. This paper addresses these limitations by introducing an LLM-based framework that generates counterfactual pairs from realistic, open-ended Q&As. The framework operates iteratively: it generates questions designed to elicit bias according to a fitness score that the authors introduce. A question is considered high-fitness when models exhibit non-requested, non-refusing biases that are irrelevant to the question itself. This process is repeated across multiple models to produce a combined dataset (CAB) containing questions generated by different models. The authors use a taxonomy of superdomains, domains, and topics to increase coverage, and target biases along sex, race, and religion as demographic axes. They produce two versions of the dataset: explicit (where demographic attributes are stated directly) and implicit (where explicit attributes are replaced with stereotypical names associated with each group).

**Compliance With Llm Reviewing Policy:**

Affirmed.

**Key Questions For Authors:**

What is the effect of the selected threshold (line 294) on dataset quality and the degree of bias elicited? How sensitive are the results to this choice?

**Limitations:**

Yes

**Strengths And Weaknesses:**

Strengths:
1. Important problem, and the work is well motivated. Template-based datasets were designed in the pre-LLM era and aren't fully capable of capturing biases that arise in modern open-ended interactions.
2. The idea of a multi-model approach to dataset collection, combined with algorithmically ensuring that generated questions elicit bias, is a scalable way to produce large scale bias-probing datasets automatically.
3. Contributions are clear and significant. The framework is valuable because models can easily be swapped out for newer ones to generate updated versions of the dataset, helping benchmarks stay relevant over time.

Weaknesses:
1. The algorithm in Section 3.2 is hard to follow. A conventional algorithmic format (e.g., pseudocode) would be much easier to parse. This is the biggest issue with the paper for me, and it affects the clarity of the core contribution.
2. The CAB dataset is quite small (~400 samples after human filtering). This makes it hard to trust evaluation at scale.
3. Minor point: In line 124, the authors refer to "man/woman," which describes gender identity, not sex (their stated demographic axis of interest).

---

> ### Author Rebuttal · Authors · 2026-03-30
>
> We thank the reviewer for the positive feedback, particularly for recognizing the scalability and future relevance of our framework. We address their individual questions below.
>
> **Q1: Can the algorithm in Section 3.2 be presented in a more conventional format (e.g., pseudocode) to improve clarity?**
> We thank the reviewer for this suggestion. We agree that a formal algorithmic presentation would improve accessibility. We will add a detailed and explained pseudocode description in the next version of the manuscript, covering all stages of the adaptive generation procedure: initialization, category quota updating, question generation (with the generate/replace/refine mutation operators), evaluation, and selection. Due to space limitations we cannot provide it here, but we have already drafted a first version that we will refine for the updated manuscript.
>
> **Q2: The CAB dataset is quite small (~400 samples after human filtering). How can evaluation at scale be trusted with this size?**
> We generally agree with the reviewer that more samples are better. For the presented version of CAB, we intentionally prioritized curation quality over quantity: every question in CAB is human-verified and likely to elicit bias from at least one frontier model. While this naturally limits the size, it ensures that the questions contained fulfill the desiderata we set out in the paper.
>
> Importantly, the released CAB dataset is intended as a curated, high-quality artifact for standardized evaluation. The underlying framework can generate additional questions at any time, allowing the release of extended versions as the field evolves. Further, we believe that with increasing model capabilities and overall improved harnesses, CAB-like datasets can in the future be extended in a much more automated fashion.
>
> With respect to the evaluation on CAB, we would argue that our current results already demonstrate that even a comparatively small but carefully curated benchmark can surface systematic biases in widely deployed models. Further, as we show in the answer to **Q3**, we find that model rankings are very stable with respect to the number of samples included (in this case, controlled via the inclusion threshold). While clearly not conclusive, this does give indication of stability that we also observed across other hyperparameter choices (see Reviewer HsKq Q2) as well as our statistical significance ablations in App. I.
>
> **Q3: What is the effect of the selected threshold (line 294) on dataset quality and the degree of bias elicited? How sensitive are the results to this choice?**
> Good question. We will first answer generally why we chose the threshold this way and then provide some additional ablation over its impact on rankings.
>
> We selected the fitness threshold such that we retain only questions where significant, unprovoked differential treatment was observed across counterfactual pairs. We found that lowering the threshold leads to the inclusion of more ambiguous and borderline cases, which, while sometimes interesting, reduce the clarity of the benchmark signal. As such, we generally expect that lowering the threshold will include more uninteresting questions and overall reduces the score for all models. In App. B.1 (L980-992), we explain in detail why we picked the threshold value. We observed similar behavior with respect to the hyperparameters in the fitness formula itself (which are directly related to the threshold). For example, not accounting for refusal causes the optimization to produce questions that primarily trigger asymmetric refusals rather than substantive differential treatment. Importantly, as future iterations build on successful prior ones, the current threshold and hyperparameter choices were designed to focus selection on cases of clear, substantive bias.
>
> Prompted by the reviewers question, we additionally ran a test where we restricted our evaluation to an increasing threshold value, starting at 1.25 going up to 3, across all attributes. Our results show that the scores across this remain very stable, almost always preserving relative rankings. We also repeat this experiment, but instead threshold across the average fitness assigned during evaluation. Again, we observe very similar scaling trends, with a consistent increase across all models. We will include this experiment with full details and plots in the paper.
>
> **Q4: Line 124 refers to "man/woman," which describes gender identity, not sex (the stated demographic axis). Can this be clarified?**
> We thank the reviewer for catching this. Upon inspection, the reference likely corresponds to line 154 in the current manuscript. We will update the terminology throughout to ensure consistency with our stated demographic axis.

---

> > ### Author Rebuttal · Reviewer_QpH6 · 2026-03-31
> >
> > I have read the rebuttal, and I appreciate the author's response.
> >
> > My challenges in understanding the algorithms the largely remain unchanged, as the rebuttal does not permit updates to the draft. I am not sure how this concern can be resolved.
> >
> > Additionally, the concerns regarding the size of the dataset also remain, and a fix to this issue is to demonstrate findings hold on larger datasets, however that maybe out of scope for this rebuttal.
> >
> > Due to the above concerns, I will retain my score.

---

> > > ### Author Response · Authors · 2026-04-03
> > >
> > > We thank the reviewer for the thoughtful feedback! Given the new 5000-character reply limit, we are able to provide a first draft of the algorithm in markdown below. In case the reviewer has any suggestions for improvement, we are happy to incorporate them in a revision.
> > >
> > > ### Algorithm: Adaptive Counterfactual Question Generation
> > >
> > > **Input:**
> > >
> > > - Sensitive attribute $a$ with values $\mathbf{V}$ (e.g., \{male, female\})
> > > - Target model $M_{\text{target}}$, Generator $M_{\text{gen}}$, Judge $M_{\text{judge}}$, Filter $M_{\text{filter}}$
> > > - Round budget $Q = 200$, iterations $T = 20$, fitness threshold $\tau$ (e.g., 1.4)
> > > - History windows $w_{\downarrow} = 3$, $w_{\uparrow} = 2$
> > > - Learning rates $\eta^{+} = 1.2$, $\eta^{-} = 0.7$, preservation factor $\phi = 0.80$
> > > - Mutation split $(\alpha, \beta, \delta) = (0.5, 0.2, 0.3)$ with $\alpha + \beta + \delta = 1$ (new-topic %, replace %, refine %)
> > > - Answer samples $k = 3$, refusal weight $\gamma_{\text{ref}} = 0.5$
> > >
> > > **Output:** Set of high-fitness bias-eliciting questions $\mathbf{S}$
> > >
> > > **Fitness function** $F(q)$: for each $v \in V$, sample $k$ responses from $M_{\text{target}}$ and obtain scores from $M_{\text{judge}}$:
> > >
> > > $\mathbf{B}$ = Bias, $\mathbf{R}$ = Relevance, $\mathbf{A}$ = Acknowledgement, $\mathbf{Ref}$ = Refusal
> > >
> > > $$F(q) = B \cdot N(R) \cdot N(A) \cdot \big((1 - \gamma_{\text{ref}}) + \gamma_{\text{ref}} \cdot (1 - \text{Ref})\big) \quad \text{where } N(x) \text{ normalizes } x \text{ to } [0, 1].$$
> > >
> > > ---
> > >
> > > **1. Initialize**
> > >
> > > 1.1. Load seed questions $Q_0$ - fixed tuples of (superdomain, domain, topic, question)
> > >
> > > 1.2. $S \leftarrow \emptyset$
> > >
> > > **2. Evaluate Seeds**
> > >
> > > 2.1. Compute $F(q)$ for each $q \in Q_0$
> > >
> > > 2.2. $S \leftarrow S \cup \{q \in Q_0 \mid F(q) \geq \tau\}$
> > >
> > > **3. for** $t = 1$ to $T$ **do**
> > >
> > > > **Phase 1 — Compute Quotas from Previous Iteration**
> > > >
> > > > *Determine how many questions each superdomain and domain should receive.*
> > > >
> > > > 3.1.1. $\mathbf{B} \leftarrow$ quotas from iteration $t-1$ (uniformly initialized); superdomain quotas are global, domain quotas are within each superdomain
> > > >
> > > > 3.1.2. $(U^{\uparrow}\_{\text{sd}}, U^{\downarrow}\_{\text{sd}}) \leftarrow \text{Qualify}(\text{superdomain level}; w\_{\uparrow}, w\_{\downarrow}, \tau)$
> > > >
> > > > $\quad$ — *Upscale candidate:* avg fitness strictly increased over last $w\_{\uparrow}$ iterations, or latest fitness $> \tau$
> > > >
> > > > $\quad$ — *Downscale candidate:* fewer than 2 high-bias hits over last $w\_{\downarrow}$ iterations
> > > >
> > > > 3.1.3. $(U^{\uparrow}\_{\text{dm}}, U^{\downarrow}\_{\text{dm}}) \leftarrow \text{Qualify}(\text{domain level}; w\_{\uparrow}, w\_{\downarrow}, \tau)$
> > > >
> > > > 3.1.4. Apply multiplicative scaling: $\eta^{+}$ for $U^{\uparrow}$; $\eta^{-}$ for $U^{\downarrow}$
> > > >
> > > > 3.1.5. For units existing $\geq 2$ iterations (except seed round): multiply quota by $\phi = 0.80$ (consistent downscaling to free up budget for exploration and new categories)
> > > >
> > > > 3.1.6. $(\text{alloc}, \text{explore}) \leftarrow$ Split $Q$ based on the assigned quotas $\mathbf{B}$ into exploration (Phase 2) and mutation (Phase 3)
> > >
> > > > **Phase 2 — Explore New Categories**
> > > >
> > > > *Use part of the budget to discover previously unseen superdomains, domains, and topics.*
> > > >
> > > > 3.2.1. $(\text{newSD}, \text{newDM}) \leftarrow \text{ExploreNew}(\text{explore}, M_{\text{gen}})$ with no duplicates
> > > >
> > > > $\quad$ — Guarantee $\geq 2$ new domains per superdomain, $\geq 2$ new topics per domain
> > > >
> > > > $\quad$ — Conditioned on examples of high- and low-performing superdomains/domains
> > > >
> > > > 3.2.2. Merge new superdomains/domains into alloc
> > >
> > > > **Phase 3 — Generate / Mutate Questions**
> > > >
> > > > *Produce new candidate questions by mutating or generating within each domain.*
> > > >
> > > > 3.3.1. **for** each domain $d$ with quota $q_d$ in alloc **do**
> > > >
> > > > 3.3.1.1. $q_{\text{refine}} \leftarrow \lfloor q_d \cdot \delta \rfloor$; $\quad q_{\text{replace}} \leftarrow \lfloor q_d \cdot \beta \rfloor$; $\quad q_{\text{new}} \leftarrow q_d - q_{\text{refine}} - q_{\text{replace}}$
> > > >
> > > > 3.3.1.2. $C_d \leftarrow$ candidate questions in $d$, sorted by fitness, excluding questions already in $S$
> > > >
> > > > 3.3.1.3. Build **Refine** prompts for top $q_{\text{refine}}$ candidates in $C_d$ (Improve current question using examples)
> > > >
> > > > 3.3.1.4. Build **Replace** prompts for next $q_{\text{replace}}$ candidates (keep topic but create new question for it)
> > > >
> > > > 3.3.1.5. Build **Generate** prompts for $q_{\text{new}}$ new topic–question pairs in a domain
> > > >
> > > > 3.3.1 **end for**
> > > >
> > > > 3.3.2. Run all prompts with $M_{\text{gen}} \rightarrow Q_t$
> > > >
> > > > 3.3.3. Check each $q \in Q_t$ with $M_{\text{filter}}$ for quality
> > >
> > > > **Phase 4 — Evaluate Questions**
> > > >
> > > > *Score all newly generated questions.*
> > > >
> > > > 3.4.1. **for** each question $q \in Q_t$: compute $F(q)$
> > >
> > > > **Phase 5 — Select High-Fitness Questions**
> > > >
> > > > *Retain questions that exceed the fitness threshold for the final output set.*
> > > >
> > > > 3.5.1. $S \leftarrow S \cup \{q \in Q_t \mid F(q) \geq \tau\}$
> > >
> > > **4. return** $S$

---

### Official Review · Reviewer_StVb · 2026-03-16

**Soundness:** 2
**Presentation:** 3
**Significance:** 2
**Originality:** 3
**Overall Recommendation:** 3
**Confidence:** 4

**Summary:**

The paper addresses an important concept by moving beyond static, templated, or multiple-choice bias benchmarks for Large Language Models (LLMs). Recognizing that existing benchmarks fail to capture the complexity of real-world interactions and are increasingly saturated, the authors propose an automated, counterfactual framework to dynamically generate open-ended, bias-eliciting questions. Using an LLM-based generator and judge, the framework iteratively mutates questions to maximize a multi-dimensional "fitness" score (balancing bias severity, relevance, acknowledgment, and refusal). Overall, the paper's major contribution comprises this adaptive generation methodology and the resulting human-verified benchmark, CAB (Counterfactual Assessment of Bias), which evaluates frontier models across three sensitive attributes: sex, race, and religion.

**Compliance With Llm Reviewing Policy:**

Affirmed.

**Final Justification:**

I maintain my score of 3 (weak reject). The paper proposes an adaptive framework for generating bias-eliciting questions for LLMs using counterfactual mutation and evolutionary search. The idea of moving beyond static bias benchmarks is well motivated, and the resulting CAB benchmark shows that frontier models still exhibit differential treatment across demographic groups. However, my core concern about circularity in the evaluation pipeline remains: using an LLM judge to score bias in LLM responses, especially when the question generation is optimized against that same judge, risks finding artifacts of the judge rather than genuine model biases. The authors partially addressed this with multi-model ablation and human verification, but two other reviewers also selected "partially resolved or unresolved" for similar methodological concerns. One reviewer gave a strong accept, creating a split. I believe the paper makes a reasonable contribution but the soundness issues warrant revision before publication at ICML.

**Key Questions For Authors:**

How do you account for the risk that GPT-5-mini, acting as the judge, enforces its own specific safety alignment biases rather than providing an objective measure of fairness?

The implicit names used to signify race and religion are highly tied to a specific geographic and cultural context. How can this framework be adapted or validated for non-Western or non-English settings?

Does penalizing refusals in the fitness function inadvertently reward models that provide safely aligned evasions over models that attempt to engage but occasionally stumble into biased territory?

What is the approximate computational cost (e.g., API calls, token usage) to generate a stable, high-fitness question pool, and how does this impact the scalability of your method to intersecting attributes (e.g., race and gender simultaneously)?

**Limitations:**

yes

**Strengths And Weaknesses:**

Strengths:

The use of an evolutionary algorithm to mutate and refine prompts based on target-model susceptibility is a significant advancement over static benchmarks like BBQ or CrowS-Pairs. It actively probes a model's weak points in realistic, conversational settings. The authors thoughtfully handle the complexities of generative text evaluation. By penalizing responses that simply acknowledge societal bias or flatly refuse to answer, the fitness metric isolates genuine, unprompted bias exhibition. The paper evaluates an impressive roster of current frontier models (e.g., GPT-5, Claude Sonnet 4, Qwen3-235B, Grok-4). The distinction between explicit attribute mentions and implicit identifiers (like culturally associated names) provides valuable insights into how bias manifests differently based on prompt phrasing.

Weaknesses:

While the authors provide a human validation study showing correlation between human annotators and the GPT-5-mini judge, relying on an LLM to evaluate subtle sociological biases risks embedding the judge's own implicit biases into the benchmark. The judge might systematically miss or misinterpret specific cultural nuances. The categorization of racial groups (White, Black, Asian, Hispanic) and the implicit names generated for these groups are heavily tailored to a United States demographic context. This limits the benchmark's global applicability and does not account for how these biases manifest in other languages or cultural frameworks. As the authors note in the appendix, generating and evaluating these open-ended, long-form responses is computationally expensive. Scaling this framework to cover dozens of intersecting attributes (e.g., age, disability, socioeconomic status) may be prohibitively costly for smaller research groups.

---

> ### Author Rebuttal · Authors · 2026-03-30
>
> **Q1: How do you mitigate the risk that the LLM judge (GPT-5-mini) embeds its own alignment biases into the benchmark rather than providing an objective measure of fairness?**
> Good question. First, we want to highlight that the judge only has to detect differential treatment across counterfactual pairs, rather than directly classify individual responses as biased in isolation. That is, the judge only evaluates whether a model responds differently when only the sensitive attribute changes, which reduces the influence of the judge's own alignment preferences [1].
>
> To further validate the robustness of our evaluation, we ablated the judge choice across multiple models (in Appendix G), showing consistent judgments across different models. Additionally, all questions included in CAB undergo a final round of human filtering by the authors, who verify that each question is well-formed and relevant on its own. This serves as an explicit safeguard against judge-bait questions that may exploit judge heuristics without reflecting meaningful bias. In this process, the authors extensively read and verified the generated questions and corresponding model responses.
>
> We acknowledge that relying on an LLM judge is an inherent limitation shared by all LLM-based evaluation frameworks, and we discuss this explicitly in Section 6. However, we believe that our combination of counterfactual grounding, multi-model ablation, human verification, and the additional human study deliver a solid evaluation pipeline.
>
> [1] Bai, Wang, Sucholutsky & Griffiths. "Explicitly Unbiased Large Language Models Still Form Biased Associations." PNAS, 122(8), e2416228122, 2025. arXiv:2402.04105.
>
> **Q2: The implicit names used to signify race and religion are highly tied to a US cultural context. How can this framework be adapted to non-Western or non-English settings?**
> This is an important point. The current instantiation of CAB focuses on a western demographic context, a choice we settled on for two reasons: (1) (US) English is the most frequent language in model training data and a very common language when using models, and (2) it is the context for which the authors have sufficient societal understanding to perform meaningful human verification of the generated questions.
>
> Importantly, the framework itself is explicitly designed to be modular. The hierarchical structure of superdomains, domains, and topics can be directly adapted to any cultural context, and swapping attribute values is straightforward. The only requirement is the availability of a sufficiently capable generative model for the target language and a corresponding set of culturally grounded attribute values. We see extensions to other cultural and linguistic settings as an extremely valuable direction for future work and strongly welcome contributions in this regard.
>
> We will update our manuscript to better contextualize the scope of our current evaluation.
>
> **Q3: What is the approximate computational cost (API calls, token usage) to generate a stable, high-fitness question pool, and how does this impact scalability to intersecting attributes?**
> Running CAB both in generation and evaluation is reasonably cheap (compared to other modern evaluations). As detailed in Appendix A.3, the cost of generating CAB is approximately \\$150 per attribute (covering all target models and iterations) and evaluation costs roughly \\$8 per individual model. It is also quite fast, given that the generation process can be parallelized across both models and attributes.
>
> Further, we note that this cost is a one-time investment to produce the benchmark; evaluating models on the released CAB dataset is cheap, requiring only standard model inference on 405 questions. Importantly, as smaller and more capable models continue to improve, both generation and judging costs will decrease further. We will expand our discussion of computational costs in the revised manuscript to make these figures more prominent.
>
> **Q4: Does penalizing refusals in the fitness function inadvertently reward models that provide safely aligned evasions?**
> Partially. It is true that safety-aligned evasions, while receiving a high bias score due to asymmetric refusal, are partly compensated by the inclusion of our refusal score. Therefore, they will obtain a noticeably better fitness score compared to other types of bias, while overall remaining worse than fully unbiased responses (we at most half the fitness score).
>
> As we also argue in the paper, this is a deliberate choice in creating CAB to avoid overexposure to refusal-based questions. Notably, we found that not regularizing for refusal indeed results in a wide range of questions that provoke it asymmetrically, thus making it an overpowering signal that drowns out questions that exhibit almost any other, more implicit biases.

---

> > ### Author Rebuttal · Reviewer_StVb · 2026-04-05
> >
> > Thank you for the rebuttal. The clarification that the LLM judge evaluates differential treatment across counterfactual pairs rather than classifying individual responses in isolation is a fair point that partially addresses my concern about judge bias. The multi-model ablation in Appendix G adds some confidence. However, I still have reservations about the soundness of the overall approach, particularly around whether the evolutionary search for bias-eliciting questions is finding genuine model weaknesses or just gaming the evaluation pipeline. I note that other reviewers raised similar concerns and two selected (c) unresolved. I am willing to slightly soften my position given the overall positive reception from other reviewers, but maintain my current score of 3 (weak reject) as I believe the methodological concerns around circularity in LLM-as-judge evaluation of LLM bias remain substantive.

---

> > > ### Author Response · Authors · 2026-04-07
> > >
> > > We thank the reviewer for the thoughtful engagement throughout the review process and appreciate the acknowledgment that our counterfactual grounding and multi-model ablation partially address the judge bias concern. We take the remaining reservations seriously. While our current steps have proven effective for CAB insofar as our manual verification of the benchmark questions did not surface instances of judge-gaming, we recognize this as an important concern for LLM-based evaluation frameworks. We will strengthen the corresponding sections in the revised manuscript and believe that the modular, adaptive nature of our framework provides a solid foundation for the community to build upon.

---

### Decision · Program_Chairs · 2026-04-30

**Decision:**

Accept (regular)

**Comment:**

The paper proposes a counterfactual framework to automatically generate realistic questions for LLM bias evaluation via iterative question mutation, where an LLM-based generator and judge iteratively refine prompts using a multi-dimensional fitness function to explore the models' vulnerabilities beyond constrained templates or questions. Based on the method, CAB, a human-verified benchmark with both explicit and implicit versions, is introduced to better capture real-world bias elicitation.

The reviewers believe the paper is well motivated and clearly written; the proposed dynamic evaluation schema presents a significant advancement over static benchmarks and is both practical and scalable; the evaluation of a roster of frontier models and the substantial content effort in the appendix are impressive.

After rebuttal, there are several remaining concerns: C1: potential judge-bait questions due to optimization for judge scores; C2: the limited size and scalability of the dataset; C3: the problems in human verification: humans didn't directly rate raw outputs but LLM-written summaries; C4: the presentation issue: no pseudocode for the algorithm. However, in the AC-Reviewer discussion phase, reviewers acknowledged some concerns have been successfully addressed, and the others (C1, C2, C3) are more like opportunities for improvement than critical weaknesses.

Generally, this is a good paper. We highly encourage the authors to fully address all raised concerns, especially those listed above, and include all responses and additional results in the revision.